# Investigation of Different Pre-Treatment Techniques and 3D Printed Turbulence Promoter to Mitigate Membrane Fouling in Dairy Wastewater Module

**DOI:** 10.3390/ma16083117

**Published:** 2023-04-15

**Authors:** Szabolcs Kertész, Aws N. Al-Tayawi, Gréta Gergely, Bence Ott, Nikolett Sz. Gulyás, Zoltán Jákói, Sándor Beszédes, Cecilia Hodúr, Tamás Szabó, Zsuzsanna László

**Affiliations:** 1Department of Biosystems Engineering, Faculty of Engineering, University of Szeged, Moszkvai Krt. 9, H-6725 Szeged, Hungary; 2Doctoral School of Environmental Sciences, University of Szeged, Tisza Lajos Krt. 103, H-6725 Szeged, Hungary; 3Faculty of Environmental Science and Technology, University of Mosul, Al-Majmoa’a Street, Mosul 41002, Iraq; 4Department of Physical Chemistry and Materials Science, University of Szeged, Rerrich Béla Tér. 1, H-6720 Szeged, Hungary

**Keywords:** membrane fouling mitigation, 3DP turbulence promoter, ultrafiltration, pre-treatments, dairy wastewater treatment

## Abstract

This study investigates the enhancement of dairy wastewater treatment using chemical and physical pre-treatments coupled with membrane separation techniques to reduce membrane fouling. Two mathematical models, namely the Hermia and resistance-in-series module, were utilized to comprehend the mechanisms of ultrafiltration (UF) membrane fouling. The predominant fouling mechanism was identified by fitting experimental data into four models. The study calculated and compared permeate flux, membrane rejection, and membrane reversible and irreversible resistance values. The gas formation was also evaluated as a post-treatment. The results showed that the pre-treatments improved UF efficiency for flux, retention, and resistance values compared to the control. Chemical pre-treatment was identified as the most effective approach to improve filtration efficiency. Physical treatments after microfiltration (MF) and UF showed better fluxes, retention, and resistance results than ultrasonic pre-treatment followed by UF. The efficacy of a three-dimensionally printed (3DP) turbulence promoter was also examined to mitigate membrane fouling. The integration of the 3DP turbulence promoter enhanced hydrodynamic conditions and increased the shear rate on the membrane surface, shortening filtration time and increasing permeate flux values. This study provides valuable insights into optimizing dairy wastewater treatment and membrane separation techniques, which can have significant implications for sustainable water resource management. The present outcomes clearly recommend the application of hybrid pre-, main- and post-treatments coupled with module-integrated turbulence promoters in dairy wastewater ultrafiltration membrane modules to increase membrane separation efficiencies.

## 1. Introduction

The increasing demand for milk and milk products has led to a substantial production of dairy wastewater during processing and cleaning operations [1]. Appropriate treatment of this wastewater is essential to prevent environmental degradation and comply with regulatory guidelines [2,3,4]. The integration of membrane-based technologies has emerged as a viable solution for the treatment of dairy wastewater [5,6]. Among these, membrane separation technologies are particularly attractive due to their efficiency, low energy consumption, and environmental friendliness [7]. However, membrane fouling remains a major challenge in the application of membrane-based technologies for dairy wastewater treatment [8]. Membrane fouling reduces process performance, shortens membrane lifespan, and increases operational costs [9,10,11,12,13,14].

Pre-treatment techniques can be applied to reduce fouling and increase permeate flux [15,16]. Physical and chemical pre-treatment methods have been found to be effective in mitigating membrane fouling and expanding the use of membrane systems beyond turbidity and pathogen reduction [17,18]. Chemical pre-treatment with “in-line” coagulation before ultrafiltration (UF) can decrease fouling by increasing the removal of organics, while enhanced flocculation pre-treatment can also prevent membrane fouling [19,20,21]. Microfiltration (MF) is an effective physical pre-treatment method for eliminating most membrane-fouling-causing contaminants from feed water, including particles, turbidity, bacteria, and large-molecular-weight organic matter [22]. Ultrasonic treatment has also emerged as a promising pre-treatment technique to prevent membrane fouling [23,24].

In addition to pretreatment methods, several effective techniques have been reported to decrease membrane fouling, including membrane surface modification through the coating [25,26] and grafting of nanoparticles onto polymeric membranes [27,28]. Among these techniques, grafting using nanoparticles is particularly advantageous due to its low operational cost and mild reaction conditions. However, these methods require the use of extra chemicals and solvents, which can be problematic from an environmental and economic perspective [29].

Recently, researchers have identified the use of feed spacers and turbulence promoters as effective methods for reducing membrane fouling. Proper design of the feed spacers and turbulence promoters can significantly decrease fouling tendencies within the membrane separation module [30]. Compared to membrane modification and pretreatment techniques, 3D printed feed elements offer a physical method that can be even more effective at reducing membrane fouling. Furthermore, 3D printed feed elements can provide a more precise surface with better characteristics, making them a focus of recent research efforts [31].

Three-dimensionally printed (3DP) turbulence promoter integration has become a promising technology for mitigating membrane fouling and enhancing filtration parameters in the membrane separation module [32,33]. In this study, we investigate the efficiency of ultrafiltration after applying chemical and physical pre-treatment, including coagulation and flocculation as chemical methods, MF and ultrasonic as physical pre-treatment methods, and 3DP turbulence promoter as a promising physical method, as part of a multi-stage dairy wastewater treatment. These processes were combined with ultrafiltration in a laboratory membrane separation device, and the permeate fluxes, resistance, and membrane rejection were examined and calculated. The effect of integrating 3DP turbulence promoter into the ultrafiltration cell was also inspected, and gas production from the concentrated part as a post-treatment was examined.

## 2. Material and Methods

### 2.1. Preparation of Dairy Wastewater Model

The dairy wastewater model was prepared by dissolving 5 g/L skimmed milk powder (InstantPack, Berettyóújfalu, Hungary) and 0.5 g/L Cl80 anionic detergent (Chemipur Cl80, Hungaro Chemicals, Nagycserkesz, Hungary) in tap water at 25 °C. The concentration was chosen based on the average pollution value of dairy wastewater (~5 g/L) reported in the literature [34]. The mixture was continuously mixed for 40 min prior to each measurement to ensure proper homogenization, and it was considered the initial dairy model wastewater.

### 2.2. Examination of Pre-Treatments before Membrane Filtration

Various physical and chemical pre-treatments were carried out to enhance the lifespan of the membranes by preventing or reducing fouling.

#### 2.2.1. Physical Pre-Treatments

To reduce turbidity (measured by Hach2100AN turbidimeter, Hach, Germany) and chemical oxygen demand (COD digester and spectrophotometer, Lovibond, Germany), microfiltration using a 0.2 µm PES membrane (Sepro, Carmel, IN, USA) was first applied as a physical pre-treatment. Bortoluzzi et al., (2017) reported that combining microfiltration and nanofiltration in wastewater treatment reduced turbidity by 100%, color by 96%, and COD by 51%. This combined pre-treatment process was more effective in retaining solids and organics than nanofiltration alone [35]. During the purification process, the retention capacity of the membrane was higher at lower pressures, allowing for the production of a cleaner filtrate. Ultrasonic treatment was also applied using a UP200S device (Ultrasonic processor Hielscher Ultrasonics GmbH, Teltow, Germany) to coagulate, aggregate, and settle colloidal particles.

#### 2.2.2. Chemical Pre-Treatments

In addition to the physical pre-treatment, chemical pre-treatment was used to reduce total phosphorus, turbidity, and COD. Two steps of chemical pre-treatment were applied: coagulation by adding 20% acetic acid (from ≥99% purity CH_3_CO_2_H, Sigma-Aldrich, Merck, Germany) to lower the pH value of the model wastewater to 4, and flocculation by adding FeCl_3_ (40 g/500 cm^3^) (from 97% purity FeCl_3_, Sigma-Aldrich, Merck, Germany) to the coagulated model wastewater to promote the precipitation and sedimentation of different matter such as proteins. Prazeres et al., (2020) demonstrated that treating cheesemaking wastewater in the dairy industry with a chemical process (FeCl_3_) reduced total phosphorus, turbidity, and chemical oxygen demand [36]. The impurities settled at the bottom as sludge, and some lighter particles floated on the surface, and impurities were removed using UF. These processes represent an opportunity to reduce industrial wastewater pollution in terms of total phosphorus, turbidity, and COD, and the treated wastewater can be used for irrigation purposes.

### 2.3. Turbulence Promoter Application and Characteristics

The best previously selected design of a single 3DP turbulence promoter was used, based on our previous work, as shown in Figure 1 [32]. The feed spacers were manufactured from polylactic acid (PLA) due to its superior properties compared to acrylonitrile butadiene styrene (ABS), as supported by the literature [37,38,39]. The promoter was manufactured using fused deposition modelling (FDM) technology, designed in Fusion 360 software (San Francisco, CA, USA, https://www.autodesk.com/products/fusion-360/overview?term=1-YEAR&tab=subscription&plc=F360, accessed on 22 June 2022) and sliced in the Ultimaker Cura 5.0.0 program (Utrecht, The Netherlands). A Creality CR-10S Pro V2 type 3D printer (Shenzhen, China) was used to print the promoter with a layer thickness of 0.2 mm, 100% fill density, a 60 °C tray, and a 215 °C printing temperature. The turbulence promoter had a size specification that included a 65 mm diameter outer layer, a smaller 39 mm diameter inner layer, a height of 14 mm, and 18 panels. The promoter was placed on the surface of the membrane using two circular rings in its bottom half. The outer ring was tightly fitted to the sealing O-ring and was immovable even when shaken. The promoter’s frame was formed by joining baffles between the circular rings, and the number and position of the baffles significantly impacted the flow conditions of the materials being separated.

### 2.4. Ultrafiltration after Laboratory Pre-Treatments

Ultrafiltration experiments were conducted using a Millipore Solvent Resistant Stirred Micro- and Ultrafiltration Cell (Merck Millipore, Darmstadt, Germany) (Figure 2) equipped with a 40 cm^2^ active membrane surface area. The device offers a rapid and effective means of concentrating smaller laboratory samples up to 300 mL at a maximum pressure of 5 bar. A polyethersulfone (PES) membrane with a molecular weight cut-off point (MWCO) of 150 kDa was used for the ultrafiltration experiments (Nadir, VSEP, Minden, LA, USA). The experiments were carried out using a constant intensive mixing speed of 400 rpm until two volume reduction ratio (VRR) had been achieved.

### 2.5. Post-Treatment

Gas production resulting from the anaerobic decomposition of the two-time ultrafiltered retentate part of dairy wastewater (VRR = 2) was measured using laboratory-scale fermenters with a total volume of 250 cm^3^. To measure the nascent absolute pressure throughout the fermentation period, the reactors were sealed with a polytetrafluoroethylene septum, and automatic manometric measuring heads (Oxi-Top IDS/B, WTW, Germany) were placed on top of the glass bottles. To ensure mesophilic conditions, a thermostatic cabinet maintained a constant temperature of 38 ± 0.2 °C. The following Equation (1) was used to calculate the biogas volume based on the registered pressure values [40]:(1)$$V_{gas}=\frac{P_{meas.}\cdot T_{norm.}\cdot V_{reac.up.}\;}{P_{atm.}\cdot T_{ferm.}}$$
where *V_gas_* is the produced gas volume [cm^3^], *P_meas._* is the measured gas pressure [Pa], *T_norm._* is the room temperature [°C], *V_reac.up._* is the reactor upper part voulme [cm^3^], *P_atm._* is the atmospheric pressure [Pa], and *T_ferm._* is the anaerob fermentation temperature [°C].

### 2.6. Membrane Performance Evaluation

Several parameters are used to assess the efficiency and performance of a membrane. Permeability, also known as flux (*J*), is a crucial parameter, indicating the volume of the filtrate that passes through the membrane per unit of time and area. Permeability can be calculated using Equation (2) [41]:(2)$$J=\frac{dV_{P}}{dt}\cdot \frac{1}{A_{M}}\;[L\cdot m^{-2}\cdot h^{-1}]$$
where *J* is the flux [L∙m^−2^∙h^−1^], *V_P_* is the permeate volume [m^3^], *A_M_* is the membrane active surface [m^2^], and *t* is the filtration time [s].

Another key parameter for evaluating membrane performance is retention (*R*), which reflects the proportion of the original solution remaining in the retentate for a particular component, and can be used to define its selectivity. The retention can be calculated using Equation (3) [42]:(3)$$R=\left(1-\frac{c_{perm}}{c_{feed}}\right)\cdot 100\;[\%]$$
where *R* is the retention percentage [%], *c_perm_* is the solution concentration in the permeate [mg∙L^−1^]. *c_feed_* is the solution concentration on the feed side [mg∙L^−1^].

The pressure difference between the two sides of the membrane was determined using Equation (4) [43]:(4)$$TMP=\frac{P_{Feed}+P_{Conc.}\;}{P_{Perm.}}-P_{Perm.}$$
where *TMP* is the transmembrane pressure [Pa], *P_Feed_* is the value of the pressure on the feed side [MPa], *P_Conc._* is the value of the pressure on the feed side measured in the case of compactions [MPa], and *P_perm._* is the value of the pressure on the permeate side [MPa].

The reduction ratio values (volume reduction ratio (*VRR*)) were determined using Equation (5) [44]:(5)$$VRR=\left(\frac{V_{in}}{V_{in}-V_{p}}\right)$$
where *V_in_* is the initial wastewater volume [m^3^], and *V_p_* is the filtration volume [m^3^].

### 2.7. Modelling

#### 2.7.1. Resistance-in-Series Model

The efficiency of the membrane separation process decreases over time, and flux values decrease accordingly. This phenomenon is attributed to concentration polarization or membrane fouling. Ideally, the resistance should be limited to the membrane resistance (*R_M_*) to avoid the formation of a polarization layer on the membrane surface and blockage of the membrane pores. The hydrodynamic resistance of the membrane was determined by measuring water flux before filtering, as shown in Equation (6) [45,46]
(6)$$R_{M}=(\frac{\mathrm{TMP}}{J_{\mathrm{WB}}\cdot \eta_{W}})[\%]$$
where *R_M_* is the membrane resistance [m^−1^], *TMP* is the transmembrane pressure [Pa], *J_WB_* is the water flux before filtration [L∙m^−2^∙h^−1^], and *η_w_* is the dynamic viscosity of water at 25 °C [Pas].

After dismantling the module and cleaning the membrane surface, pore fouling-induced resistance values (*R_IRREV_*, *R_REV_*) can be determined using Equations (7) and (8) [46]:(7)$$R_{\mathrm{IRREV}}=\left(\frac{TMP}{J_{WA\cdot }\eta_{W}}-R_{M}\right)\;[\%]$$
(8)$$R_{\mathrm{REV}}=\left(\frac{TMP}{J_{WA\cdot }\eta_{W}}-R_{M}-R_{IRREV}\right)\left[\%\right]\;\;$$
where *R_IRREV_* is the irreversible resistance [m^−1^], *J_WA_* is the water flux of the membrane after filtration [L∙m^−2^∙h^−1^], *η_W_* is the dynamic viscosity of water at 25 °C [Pas], *R_REV_* is the reversible resistance [m^−1^], and *R_T_* is the total resistance [m^−1^].

The total resistance (*R_T_*) consists of membrane e resistance (*R_M_*), reversible resistance (*R_REV_*), and irreversible resistance (*R_IRREV_*), which can be calculated with Equation (9), [47,48]
*R_T_ = R_M_ + R_IRREV_ + R_REV_* [%](9)

#### 2.7.2. Hermia Module

Membrane fouling phenomena can be classified, explained, and analyzed using semi-empirical and empirical mathematical models, such as the Hermia module. Hermia (1982) developed a semi-empirical mathematical model to decrease permeate flux. The Hermia model employs a typical constant pressure filtering approach, and several studies use it to determine membrane occlusion. The Hermia model includes several blocking models, such as complete blocking model, standard blocking model, intermediate blocking model, and cake layer formation model [49,50] (Figure 3).

## 3. Results and Discussion

### 3.1. Examination of Flux Changes

The dairy wastewater model measurements were conducted at room temperature and a constant pressure of 0.4 TMP to investigate permeate flux improvements. Initially, membrane separations were performed with and without stirring. The resulting permeate fluxes were examined over time (Figure 4), which indicated an initial rapid decline in fluxes (up to approximately 500 s), followed by a slower decrease that remained separate for both cases. The initial decline in flux was attributed to membrane fouling, concentration polarization, and the development of hydrodynamic flow conditions. Subsequent measurements revealed that intensive mixing at 400 rpm led to higher flux values at all measurement points. The volume reduction ratio was also doubled in about half the time compared to the non-mixed case (2899 s instead of 6338 s). Consequently, intensive stirring led to higher flux values and shorter filtration times. Therefore, subsequent experiments were carried out with mixing at 400 rpm.

### 3.2. Examination of the Different Pre-Treatments

Prior to measuring membrane performance, various pre-treatment techniques were investigated to determine the most efficient procedure for subsequent ultrafiltration (UF) (see Figure 5). The results demonstrate that UF produced superior, higher flux values compared to the control UF experiments (with and without mixing), regardless of the pre-treatment employed. Among the pre-treatment methods investigated, chemical pre-treatment was found to be the most effective. Specifically, both pre-treatment coagulation and coagulation + flocculation followed by UF yielded significantly higher fluxes and reduced total filtration time. Moreover, both physical pre-treatments, microfiltration (MF) and ultrasonic (US) reduced the filtration time. Importantly, the results highlight that MF was an excellent physical pre-treatment compared to the US treatment in terms of flux.

### 3.3. Comparison of Hermia Model Calculations

During the experimental investigation, Hermia models including the complete blocking, standard blocking, intermediate blocking, and cake layer models were employed, as depicted in Figure 6 in the case of ultrafiltration after coagulation experiment. Subsequently, the simulated results were compared to the experimental data, revealing that the cake layer model provided the most accurate representation of the membrane separation process.

The present study demonstrated that utilizing a 3DP turbulence promoter integration approach coupled with a stirring speed of 400 rpm in the membrane module can enhance the overall efficiency of the membrane separation process. The flux values obtained from the cake layer model with and without 3DP turbulence promoter are presented in (Figure 7). The results indicate that the calculated and measured values exhibit a close agreement, indicating that the cake layer model can adequately account for membrane fouling. The difference between the calculated and measured flux values with and without 3DP turbulence promoter is not significant, suggesting that the utilization of 3DP turbulence promoter does not produce a substantial impact on the flux behavior.

### 3.4. Evaluation of Membrane Retention

The retention of organic matter by the membrane was determined by performing analytical measurements on the initial wastewater and the remaining permeate after filtration. The membrane’s retention values for COD organic matter were calculated using Equation (3). Figure 8 depicts the change in retention values of pre-treatment ultrafiltration with respect to the organic matter content, presented as COD.

During individual measurements, the average COD value of the initial dairy wastewater model was 5000 mg/L. The results indicate that the pre-treatment methods substantially improved membrane retention values in this case. The enhancements were approximately 10%, 62%, 51%, 68%, and 50% for mixing, MF, coagulation, coagulation + flocculation, and ultrasonic methods, respectively, compared to the control measurement.

Additionally, the pH and conductivity of the samples were measured. The pH ranged between 7.5 and 8.5 during the measurements. Minimal differences were observed between the permeates of the ultrafiltration and the initial dairy wastewater model. For chemical pre-treatments, the pH was adjusted to 4, which did not significantly impact the measurement results. The specific conductivity was also evaluated and ranged between 800 and 1100 µS/cm. The filtrate displayed a 10% lower value than the concentrate, in which the specific conductivity values were 10% higher. This variation can be attributed to the salt content, which affects measurement results; a significant portion of salts passes through the pores of ultrafiltration membranes with a cut-off value of 150 kDa, which may explain the relatively low retention rates.

### 3.5. Membrane Resistances

To investigate the mechanisms responsible for membrane fouling, we calculated the resistance values using Equations (6)–(9) and examined their changes. Figure 9 shows the reversible and irreversible resistance values of the membrane. The total resistance values were influenced by the pre-treatments and could be expressed as a percentage change. Chemical pre-treatments, such as coagulation and coagulation plus flocculation, caused the most significant reduction in resistance values by 88.88% and 89.47%, respectively. Physical pre-treatments, such as MF and US, resulted in a decrease in resistance values by 66.72% and 33.4%, respectively, while smooth mixing caused a 50% reduction.

The figure clearly indicates that the reversible resistance values were the highest in all cases, indicating that the cake layer model had formed, which is consistent with the values calculated using the Hermia model. This could be attributed to the formation of a cake layer that hindered the free flow of the flux by reducing the number of particles and molecules entering the inner pores of the membrane, thereby adhering to the inner walls. The control ultrafiltration without stirring resulted in the lowest fluxes and the highest retention values, with the highest resistances, as indicated in the figure.

Table 1 provides information on the distribution of the membrane’s resistances and the ratios of irreversible and reversible resistances within the total resistances. The type of pre-treatment used had a significant impact on the membrane’s resistance, with irreversible resistance values ranging from 3.38% to 30.67% and reversible resistance values ranging from 30.99% to 93.47%. The smallest irreversible and the largest reversible resistance values were observed in the case of the control measurements, while the highest irreversible and the lowest reversible resistance values were observed in the case of coagulation/UF. This result could be attributed to the polarization layer becoming more prominent in the coagulation/UF case, which is easily removed and results in a lower resistance value. In contrast, the more difficult to remove, more compact polarization layer had lower resistance values, indicating the presence of the cake layer model.

Figure 10 illustrates the effect of integrating a 3D printed (3DP) turbulence promoter into a filtration cell on resistance values. Reversible resistance was identified as a key component of the resistance, implying that membrane fouling occurred based on the cake layer model, which is more easily removable than irreversible fouling. Filtration tests conducted without the 3DP turbulence promoter showed maximum total resistance (R_T_) values at a mixing speed of 0 rpm, which decreased considerably with increasing speed. However, incorporating the 3DP turbulence promoter into the cell resulted in a more pronounced decrease in total resistance values, with reductions of up to 78% observed at a mixing speed of 400 rpm. These results indicate that using the 3DP turbulence promoter increases the shear rate at the membrane surface, leading to more favorable flux values under altered flow conditions. These findings are consistent with those reported by Ferreira et al., (2020), who observed a 78% increase in permeate flux when using a 3DP turbulence promoter [51].

### 3.6. Results of Post-Treatment Gas Formation Determination

After completing the ultrafiltration (UF) experiments, 50 cm^3^ of the concentrated samples were analyzed to quantify the gas produced from them. The results show that during the UF after microfiltration treatment, no gas formation was observed (Figure 11). This can be attributed to the fact that the microfilter membrane retained most of the proteins, thereby limiting the growth and adaptation of microbial strains involved in the hydrolytic and methanogenic sub-processes. Conversely, the absence of proteins negatively affected the gas formation of the coagulation-only chemical pre-treatment. In this experiment, the low pH of 4 denatured the proteins, resulting in insignificant gas formation. The ultrasound pre-treatment experiment yielded results similar to those of the control measurement. However, on the 10th day of fermentation, the rate of gas formation increased significantly. This can be attributed to the fact that the cavitation induced by the ultrasonic pre-treatment altered the physicochemical structure of different substances, increasing the soluble organic content and thus enhancing biodegradability. However, since no substance was added to promote gas formation, the total amount of gas produced did not increase during the examined period. The most promising result was observed for the FeCl_3_ chemical pre-treatment, where ferric ions released from FeCl_3_ were found to increase the metabolic production of specific methanogenic bacterial strains, thereby promoting an increase in gas formation [52].

## 4. Conclusions

The present study investigated the efficiency of dairy wastewater treatment through different chemical and physical pre-treatments followed by ultrafiltration (UF). The Hermia and resistance-in-series models were utilized to optimize process parameters and investigate the dominant fouling mechanism. The results indicated that UF with pre-treatment was more effective than the control UF experiments without pre-treatment. Coagulation and coagulation + flocculation followed by UF were found to be the most effective chemical pre-treatments, while microfiltration outperformed the ultrasonic treatment type among the physical pre-treatments. Chemical pre-treatments reduced the resistance values the most, with the Hermia model accurately predicting the dominant fouling process as the cake layer model. The study also found that FeCl_3_ chemical treatment produced the best results in terms of gas production, which was used as a quantifiable value to evaluate the treatment efficiency. Finally, incorporating a three-dimensionally printed (3DP) turbulence promoter into the membrane module improved the efficiency of the membrane process by increasing the permeate flux and decreasing the resistance, which mitigated membrane fouling. Overall, this study demonstrates that the combination of pre-treatment methods followed by UF can significantly enhance the efficiency of dairy wastewater treatment. The results provide valuable insights into the fouling mechanisms and the optimization of process parameters, which can be beneficial for the development of sustainable wastewater treatment technologies.

## Figures and Tables

**Figure 1 materials-16-03117-f001:**
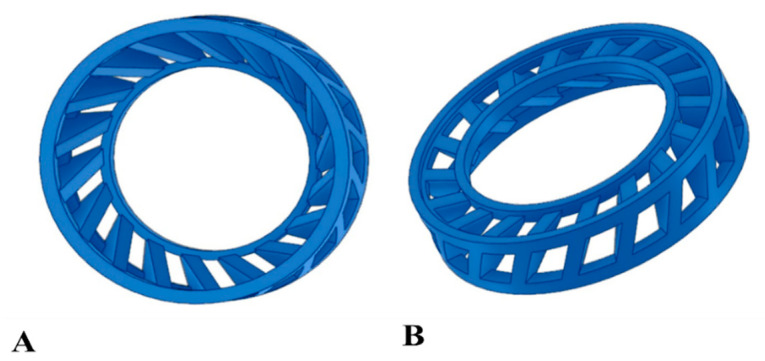
The tested 3DP turbulence promoter, (**A**) Front view and (**B**) bottom-side view.

**Figure 2 materials-16-03117-f002:**
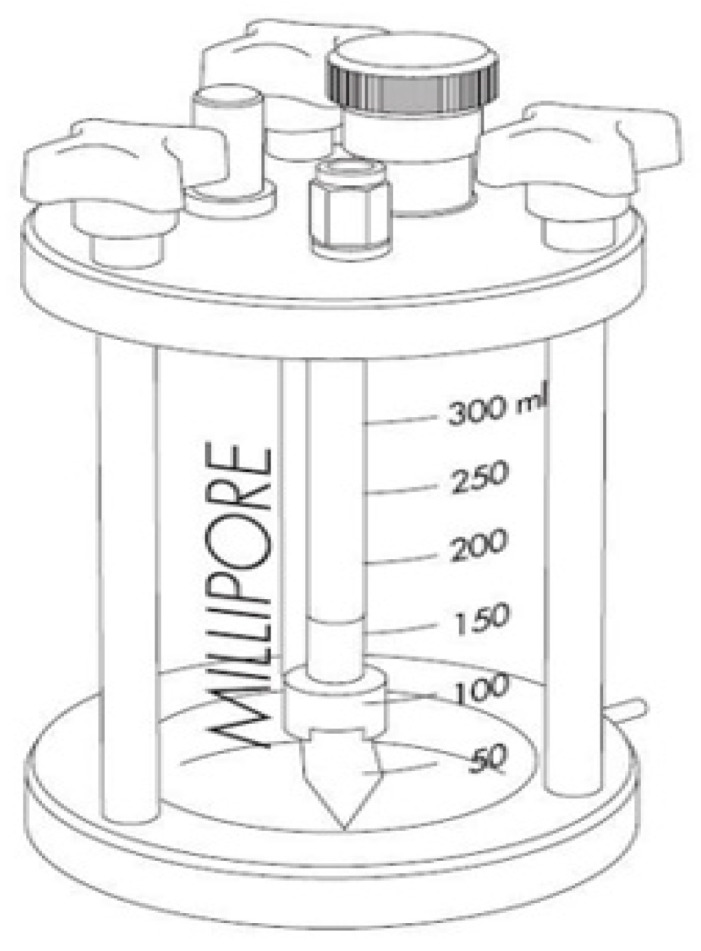
Schematic diagram of the pressure filter used (from device book).

**Figure 3 materials-16-03117-f003:**
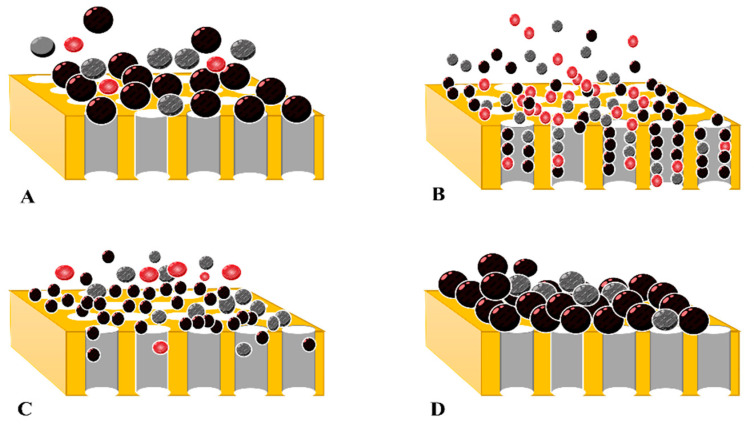
Types of fouling mechanisms according to Hermia module (**A**) complete blocking model; (**B**) standard blocking model; (**C**) intermediate blocking model; (**D**) cake layer formation model.

**Figure 4 materials-16-03117-f004:**
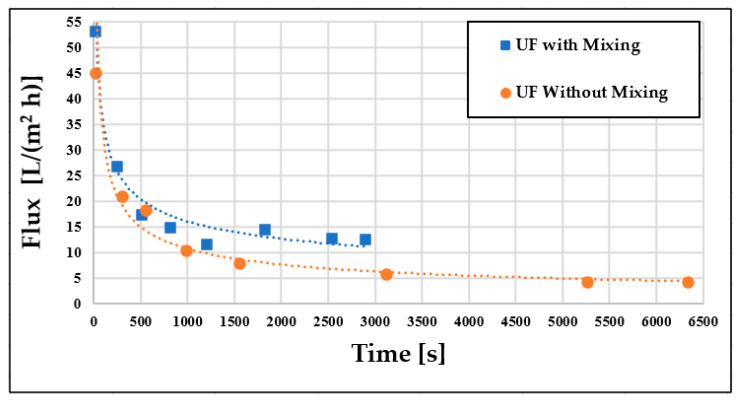
Flux changes over time without pre-treatment (T = 25 °C; TMP = 0.25 bar).

**Figure 5 materials-16-03117-f005:**
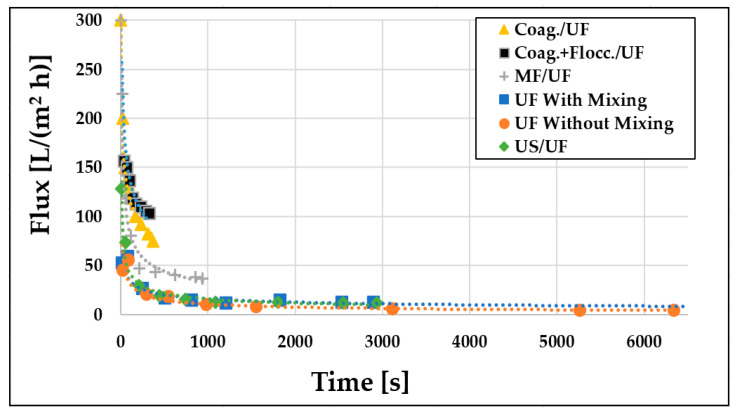
Ultrafiltration permeate flux values of 150 kDa UF membrane as a function of filtration time after different pre-treatments (T = 25 °C; TMP = 0.25 bar; *n* = 400 rpm).

**Figure 6 materials-16-03117-f006:**
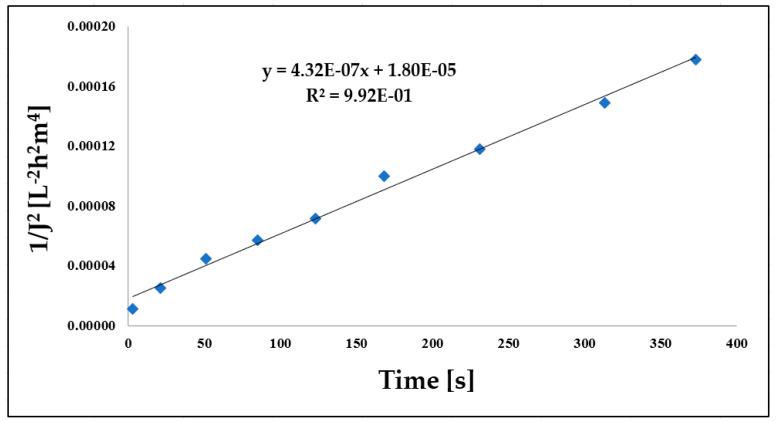
Accuracy of matching values calculated with the cake layer model equation (Coag./UF) (T = 25 °C; TMP = 0.25 bar).

**Figure 7 materials-16-03117-f007:**
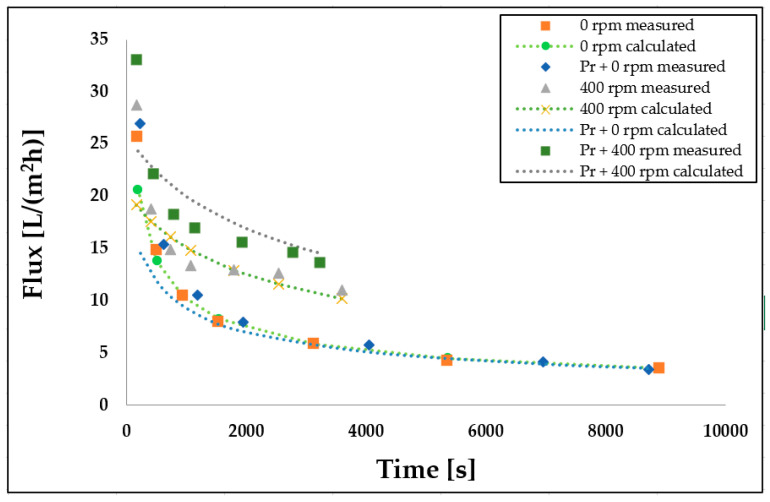
Changes in the permeate flux values of the 3DP turbulence promoter measurements as a function of time (T = 25 °C; TMP = 0.25 bar).

**Figure 8 materials-16-03117-f008:**
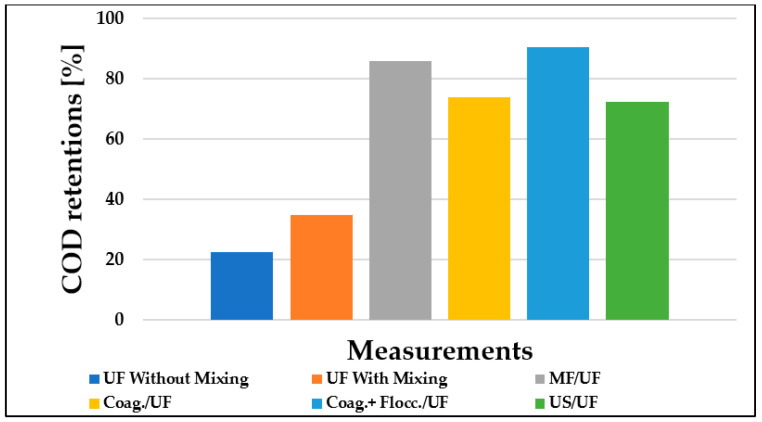
Ultrafiltration chemical oxygen demand retention values of dairy wastewater using different pre-treatments (150 kDa) (T = 25 °C; TMP = 0.25 bar).

**Figure 9 materials-16-03117-f009:**
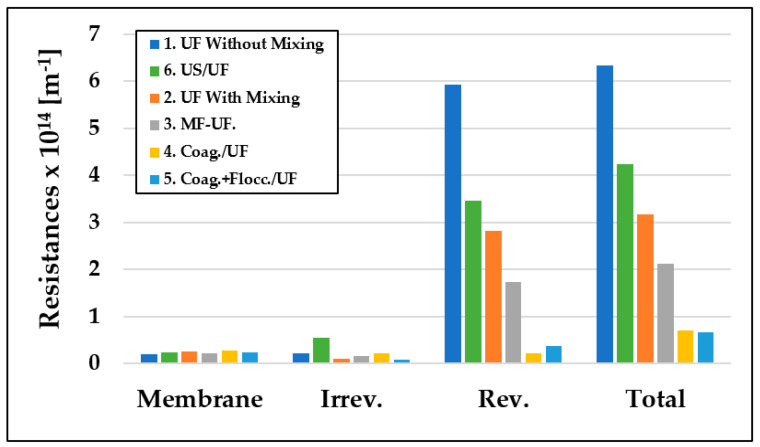
Changes in membrane resistance values due to different pre-treatment methods (T = 25 °C; TMP = 0.25 bar).

**Figure 10 materials-16-03117-f010:**
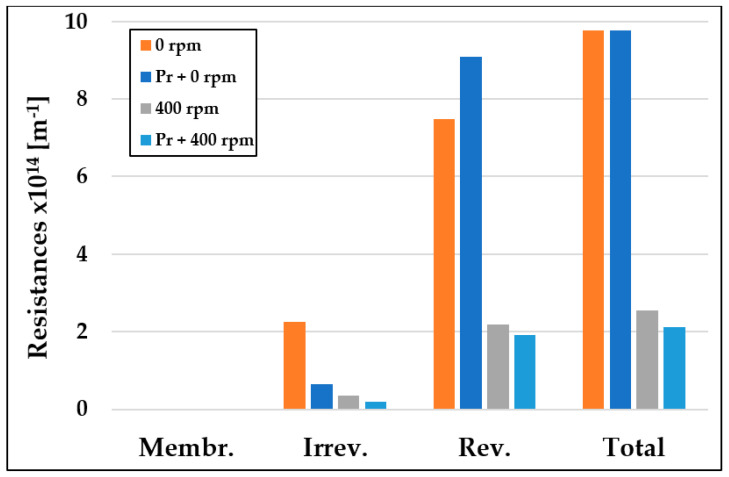
Changes in membrane resistance values due to integrating 3DP turbulence promoter and high mixing speed of 400 mixing velocity (T = 25 °C; TMP = 0.25 bar).

**Figure 11 materials-16-03117-f011:**
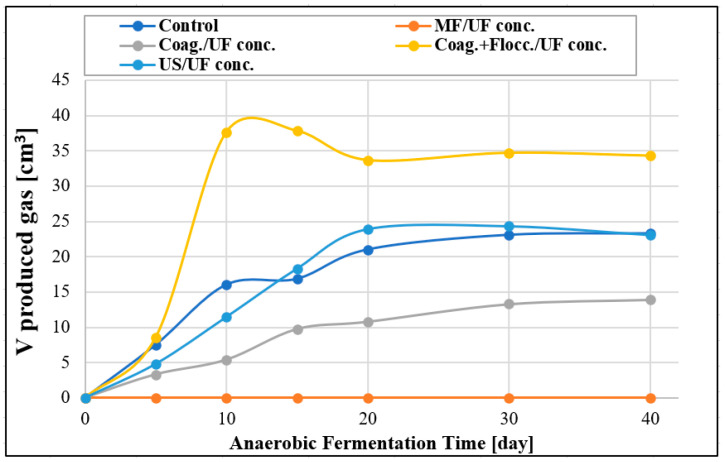
Examination of gas formations of samples with anaerobic fermentation.

**Table 1 materials-16-03117-t001:** Membrane resistances in percentage distribution.

Measurements	Distribution of Resistance Values				
	*R_m_*	*R_irrev_*	*R_rev_*	*R_irrev_/_Rrev_*	*R_total_*
Coag./UF	38.33	30.67	30.99	0.9896	100.00
Coag.+Flocc./UF	35.07	10.79	54.14	0.1992	100.00
US/UF	5.54	12.78	81.68	0.1564	100.00
MF/UF	10.22	7.67	82.11	0.0934	100.00
UF with mixing	7.95	3.41	88.64	0.0384	100.00
UF without mixing	3.15	3.38	93.47	0.0361	100.00

## Data Availability

The data are contained within the article.

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
