# Peer review of "Investigation of Different Pre-Treatment Techniques and 3D Printed Turbulence Promoter to Mitigate Membrane Fouling in Dairy Wastewater Module"

_materials, 2023, doi:10.3390/ma16083117_

Round 1

Reviewer 1 Report

This research investigated the enhancement of dairy wastewater treatment using chemical and physical pre-treatments coupled with membrane separation techniques to reduce membrane fouling. Mathematical models were utilized to comprehend the mechanisms of membrane fouling. The predominant fouling mechanism was identified by fitting experimental data into models. The results showed that the pre-treatments improved membrane efficiency for flux, retention and resistance. This study provides valuable insights into the optimization of dairy wastewater treatment and membrane separation techniques. This draft needs to address the following concerns to improve its quality.

1. In the introduction, the authors talk about pre-treatment techniques, then shared the “three-dimensional printed turbulence promoter integration”, which cannot belong to the pre-treatment method. Does this method have strong benefits compared with other methods? More ways that can reduce or prevent from membrane fouling should be added (like surface modification), then discuss their advantages and disadvantages.

2. Line 111, the authors told “the best previously selected design of a single 3DP turbulence promoter was used, as shown in (Figure 1)”. Please give the reason why it is the best selected design.

3. Please check your draft give further information about your chemical reagents, including provider and purity.

4. Line 129, “The experiments were carried out using a constant intensive mixing speed of 400 rpm until a compression ratio of 2x was reached”, maybe this sentence means that you want to increases the concentration to twice the initial level? Please give the understandable description.

5. Please explain equation 2, equation 4 and equation 5 by detail and add the related references.

6. Check the draft and add the descriptions of subscripts appearing in the formula.

7. Clearly gives equipment and methods required to characterize the substances in solution.

Author Response

Dear Reviewer,

Many thanks for your accurate feedback, which certainly has enhanced the quality of the manuscript. We have acted closely on your feedback by:

The new version contains the following changes according to your general comments

According to your specific comments and suggestions

  1. In the introduction, the authors talk about pre-treatment techniques, then shared the “three-dimensional printed turbulence promoter integration”, which cannot belong to the pre-treatment method. Does this method have strong benefits compared with other methods? More ways that can reduce or prevent from membrane fouling should be added (like surface modification), then discuss their advantages and disadvantages.

Because of your suggestion, we added the following paragraph into the Introduction part with 7 more article citations:

“In addition to pretreatment methods, several effective techniques have been reported to decrease membrane fouling, including membrane surface modification by coating [25,26] and grafting of nanoparticles onto polymeric membranes [27,28]. Among these techniques, grafting using nanoparticles is particularly advantageous due to its low operational cost and mild reaction conditions. However, these methods require the use of extra chemicals and solvents, which can be problematic from an environmental and economic perspective [29].

Recently, researchers have identified the use of feed spacers and turbulence promoters as effective methods for reducing membrane fouling. Proper design of the feed spacers and turbulence promoters can significantly decrease fouling tendencies within the membrane separation module [30]. Compared to membrane modification and pretreatment techniques, 3D printed feed elements offer a physical method that can be even more effective at reducing membrane fouling. Furthermore, 3D printed feed elements can provide a more precise surface with better characteristics, making them a focus of recent research efforts [31].”

  1. Line 111, the authors told “the best previously selected design of a single 3DP turbulence promoter was used, as shown in (Figure 1)”. Please give the reason why it is the best selected design.

Because of your suggestion, we modified the mentioned sentence to this:

“The best previously selected design of a single 3DP turbulence promoter was used, based on our previous work, as shown in (Figure 1) [32].”

which we reported that the best performance was recorded using the third turbulence promoter type (Pr. 3.) with a 400-rpm stirring speed regarding the results of fluxes, membrane resistances, and COD rejection.

  1. Please check your draft give further information about your chemical reagents, including provider and purity.

Please find the revised manuscript!

Based on your suggestion, we specified the following materials and their characteristics in the Materials and Methods:

“The dairy wastewater model was prepared by dissolving 5 g/L skimmed milk powder (InstantPack, Berettyóújfalu, Hungary) and 0.5 g/L Cl80 anionic detergent (Chemipur Cl80, Hungaro Chemicals, Nagycserkesz, Hungary) in tap water at 25 ℃.”

“To reduce turbidity (measured by Hach2100AN turbidimeter, Hach, Germany) and chemical oxygen demand (COD digester and spectrophotometer, Lovibond, Germany), microfiltration using a 0.2 µm PES membrane (Sepro, USA) was first applied as a physical pre-treatment.”

“In addition to physical pre-treatment, chemical pre-treatment was used to reduce total phosphorus, turbidity, and COD. Two steps of chemical pre-treatment were applied: coagulation by adding 20% acetic acid (from ≥99% purity CH3CO2H, Sigma-Aldrich, Merck, Germany) to lower the pH value of the model wastewater to 4, and flocculation by adding FeCl3 (40g/500 cm3) (from 97% purity FeCl3, Sigma-Aldrich, Merck, Germany) to the coagulated model wastewater to promote the precipitation and sedimentation of different matter such as proteins.”

“A polyethersulfone (PES) membrane with a molecular weight cut-off (MWCO) of 150 kDa was used for ultrafiltration experiments (Nadir, USA).”

“To measure the nascent absolute pressure throughout the fermentation period, the reactors were sealed with a polytetrafluoroethylene septum, and automatic manometric measuring heads (Oxi-Top IDS/B, WTW, Germany) were placed on top of the glass bottles.”

  1. Line 129, “The experiments were carried out using a constant intensive mixing speed of 400 rpm until a compression ratio of 2x was reached”, maybe this sentence means that you want to increases the concentration to twice the initial level? Please give the understandable description.

Yes, based on your suggestion we corrugated the mentioned sentences to the following:

“The experiments were carried out using a constant intensive mixing speed of 400 rpm until two Volume reduction ratio (VRR).”

We think that now it is more understandable. 50 ml of permeate volume was collected from the original feed volume of 100 ml during all ultrafiltration experiments. We always finish the experiments, when we reached this VRR=2 for the right comparability.

  1. Please explain equation 2, equation 4 and equation 5 by detail and add the related references.

Thanks, yes, based on your suggestion we added the sources of the mentioned equations:

“Several parameters are used to assess the efficiency and performance of a membrane. Permeability, also known as flux (J), is a crucial parameter, indicating the volume of filtrate that passes through the membrane per unit of time and area. Permeability can be calculated using Equation (2) [41]:

J [Lm-2 h-1]

(2)

Where J is the flux [L∙m-2∙h-1], VP is the permeate volume [m3], AM is the membrane active surface [m2], and t is the filtration time [s].

Another key parameter for evaluating membrane performance is retention (R), which reflects the proportion of the original solution remaining in the retentate for a particular component, and can be used to define its selectivity. The retention can be calculated using Equation (3) [42]:

[%]

(3)

Where R is the retention percentage [%], cperm is the solution concentration in the permeate [mg∙L-1]. cfeed is the solution concentration on the feed side [mg∙L-1].

The pressure difference between the two sides of the membrane was determined using Equation (4) [43]:

(4)

Where TMP is the transmembrane pressure [Pa], PFeed is the value of the pressure on the feed side [MPa], PConc. is the value of the pressure on the feed side measured in the case of compactions [MPa], and Pperm. is the value of the pressure on the permeate side [MPa].”

The reduction ratio values (Volume Reduction Ratio (VRR)) were determined using Equation (5) [44]:

(5)

Where Vin  is the initial wastewater volume [m3], and Vp is the filtration volume [m3].

  1. Check the draft and add the descriptions of subscripts appearing in the formula.

Thank you for this feedback, we checked and add the descriptions of subscripts appearing in the formula.

  1. Clearly gives equipment and methods required to characterize the substances in solution.

Thank you for this suggestion!

Please find the revised manuscript!

Based on your suggestion, we specified the physical, chemical pre-treatment methods and analytics in more details in the Materials and Methods. We rewrote the 2.2.1, 2.2.2, 2.3 and 2.4.-2.5.-2.6. sections.

And please find the attached newer version of our manuscript.

Reviewer 2 Report

1. If authors can revise the title, an attractive title can fit the manuscript

2. Add some previous and relevant studies in the introduction and in the results/discussion too if available.

3. Improve the abstract section by pointing out ts and applications.

4. There are some technical errors and grammatical errors in the manuscript. Please give detailed revisions on those parts to improve the overall language of the manuscript.

5. Follow the units and symbols as per the journal's regulations.

6. morphology of the prepared material needs to be explained with more clarity.

7. Chemicals and materials used in this experiment were not discussed in detail. Authors are suggested to include them along with purity and purchase details.

Author Response

Dear Reviewer,

Thanks for your accurate feedback, which certainly has enhanced the quality of the manuscript. We have acted closely on your feedback by:

The new version contains the following changes according to your general comments;

Improved the article overall (from the abstract to the references) which described in the following table:

N

General comments

The answers

1

Does the introduction provide sufficient background and include all relevant references?

Must be improved

Paraphrased some sentences in the Introduction and added 13 relevant new citations [25-31] for introduction and [34-39] for Materials and Methods.

2

Are all the cited references relevant to the research?

Can be improved

Because of your suggestion we checked them again one by one and we really feel that all of them are relevant, and we added some more real prevalent as we mention in the previous point.

3

Is the research design appropriate?

Can be improved

The manuscript was checked and redesigned by rewritten some part, both Materials and Methods and results and discussion.

4

Are the methods adequately described?

Can be improved

We specified the materials (chemical reagents) and their characteristics for Materials and Methods, including provider and purity. Moreover, we added more details about the used turbulence promoter and three relevant, supported citations.

5

Are the results clearly presented?

Must be improved

Regarding the Results and Discussion part, many sentences were paraphrased and improved.

6

Are the conclusions supported by the results?

Must be improved

Finally, we have rewritten the conclusion almost totally.

According to your specific comments and suggestions

  1. If authors can revise the title, an attractive title can fit the manuscript

Based on your suggestion, we can reverse the title to the following “Mitigating Membrane Fouling in Dairy Wastewater Treatment: Investigation of Pre-treatment Techniques and the Use of 3D Printed Turbulence Promoters

  1. Add some previous and relevant studies in the introduction and in the results/discussion too if available.

Because of your suggestion, we added the following paragraph into the Introduction part with 10 more article citations (also an other 3 citations were added into the Materials and Methods):

“In addition to pretreatment methods, several effective techniques have been reported to decrease membrane fouling, including membrane surface modification by coating [25,26] and grafting of nanoparticles onto polymeric membranes [27,28]. Among these techniques, grafting using nanoparticles is particularly advantageous due to its low operational cost and mild reaction conditions. However, these methods require the use of extra chemicals and solvents, which can be problematic from an environmental and economic perspective [29].

Recently, researchers have identified the use of feed spacers and turbulence promoters as effective methods for reducing membrane fouling. Proper design of the feed spacers and turbulence promoters can significantly decrease fouling tendencies within the membrane separation module [30]. Compared to membrane modification and pretreatment techniques, 3D printed feed elements offer a physical method that can be even more effective at reducing membrane fouling. Furthermore, 3D printed feed elements can provide a more precise surface with better characteristics, making them a focus of recent research efforts [31].”

  1. Improve the abstract section by pointing out ts and applications.

Because of your suggestion, we added an extra sentence to the Abstract part:

“The present outcomes clearly recommend the application of hybrid: pre-, main- and post-treatments coupled with module-integrated turbulence promoters in dairy wastewater ultrafiltration membrane modules to increase membrane separation efficiencies.”

Also we have same small changes to improve the quality of it, so the final version of our abstract is:

“This study investigates the enhancement of dairy wastewater treatment using chemical and physical pre-treatments coupled with membrane separation techniques to reduce membrane fouling. Two mathematical models, namely the Hermia and resistance-in-series module, were utilized to comprehend the mechanisms of ultrafiltration (UF) membrane fouling. The predominant fouling mechanism was identified by fitting experimental data into four models. The study calculated and compared permeate flux, membrane rejection, and membrane reversible and irreversible resistance values. The gas formation was also evaluated as a post-treatment. The results showed that the pre-treatments improved UF efficiency for flux, retention, and resistance values compared to the control. Chemical pre-treatment was identified as the most effective approach to improving filtration efficiency. Physical treatments after microfiltration (MF) and UF showed better fluxes, retention, and resistance results than ultrasonic pre-treatment followed by UF. The efficacy of a Three-Dimensional Printed (3DP) turbulence promoter was also examined to mitigate membrane fouling. Integration of 3DP turbulence promoter enhanced hydrodynamic conditions and increased shear rate on the membrane surface, shortening filtration time and increasing permeate flux values. This study provides valuable insights into optimizing dairy wastewater treatment and membrane separation techniques, which can have significant implications for sustainable water resource management. The present outcomes clearly recommend the application of hybrid: pre-, main- and post-treatments coupled with module-integrated turbulence promoters in dairy wastewater ultrafiltration membrane modules to increase membrane separation efficiencies.”

  1. There are some technical errors and grammatical errors in the manuscript. Please give detailed revisions on those parts to improve the overall language of the manuscript.

Thanks, yes, based on your suggestion we checked and corrected the errors and improved the overall language of the manuscript.

  1. Follow the units and symbols as per the journal's regulations.

Thank you for this feedback, we checked and followed the units and symbols as per the journal's regulations.

  1. morphology of the prepared material needs to be explained with more clarity.

We have explained the morphology of the material in more details and also we add  specific about the turbulence promoter, as you can see in the section (2.3 Turbulence promoter application and characteristics, line 5):

“The best previously selected design of a single 3DP turbulence promoter was used, based on our previous work, as shown in (Figure 1) [32]. The feed spacers were manufactured from polylactic acid (PLA) due to its superior properties compared to acrylonitrile butadiene styrene (ABS), as supported by literature [37–39]. The promoter was manufactured using Fused Deposition Modelling (FDM) technology, designed in Fusion 360 software and sliced in the Ultimate Cure 5.0.0 program. A Creality CR-10S Pro V2 type 3D printer (China) was used to print the promoter with a layer thickness of 0.2 mm, 100% fill density, a 60°C tray, and a 215°C printing temperature. The turbulence promoter had a size specification that included a 65 mm diameter outer layer, a smaller 39 mm diameter inner layer, a height of 14 mm, and 18 panels. The promoter was placed on the surface of the membrane using two circular rings in its bottom half. The outer ring was tightly fitted to the sealing O-ring and was immovable even when shaken. The promoter's frame was formed by joining baffles between the circular rings, and the number and position of the baffles significantly impacted the flow conditions of the materials being separated.”

  1. Chemicals and materials used in this experiment were not discussed in detail. Authors are suggested to include them along with purity and purchase details.

Based on your suggestion, we specified the following materials and their characteristics in the Materials and Methods:

“The dairy wastewater model was prepared by dissolving 5 g/L skimmed milk powder (InstantPack, Berettyóújfalu, Hungary) and 0.5 g/L Cl80 anionic detergent (Chemipur Cl80, Hungaro Chemicals, Nagycserkesz, Hungary) in tap water at 25 ℃.”

“To reduce turbidity (measured by Hach2100AN turbidimeter, Hach, Germany) and chemical oxygen demand (COD digester and spectrophotometer, Lovibond, Germany), microfiltration using a 0.2 µm PES membrane (Sepro, USA) was first applied as a physical pre-treatment.”

“In addition to physical pre-treatment, chemical pre-treatment was used to reduce total phosphorus, turbidity, and COD. Two steps of chemical pre-treatment were applied: coagulation by adding 20% acetic acid (from ≥99% purity CH3CO2H, Sigma-Aldrich, Merck, Germany) to lower the pH value of the model wastewater to 4, and flocculation by adding FeCl3 (40g/500 cm3) (from 97% purity FeCl3, Sigma-Aldrich, Merck, Germany) to the coagulated model wastewater to promote the precipitation and sedimentation of different matter such as proteins.”

“A polyethersulfone (PES) membrane with a molecular weight cut-off (MWCO) of 150 kDa was used for ultrafiltration experiments (Nadir, USA).”

“To measure the nascent absolute pressure throughout the fermentation period, the reactors were sealed with a polytetrafluoroethylene septum, and automatic manometric measuring heads (Oxi-Top IDS/B, WTW, Germany) were placed on top of the glass bottles.”

And please find the attached newer version of our manuscript.